# Establishment of Translational Luciferase-Based Cancer Models to Evaluate Antitumoral Therapies

**DOI:** 10.3390/ijms251910418

**Published:** 2024-09-27

**Authors:** Martin R. Ramos-Gonzalez, Nagabhishek Sirpu Natesh, Satyanarayana Rachagani, James Amos-Landgraf, Haval Shirwan, Esma S. Yolcu, Jorge G. Gomez-Gutierrez

**Affiliations:** 1Roy Blunt NextGen Precision Health Institute, University of Missouri, Columbia, MO 65211, USA; mr.ramos@missouri.edu (M.R.R.-G.); snagabhishek@missouri.edu (N.S.N.); srachagani@missouri.edu (S.R.); haval.shirwan@health.missouri.edu (H.S.); esma.yolcu@health.missouri.edu (E.S.Y.); 2Ellis Fischel Cancer Center, School of Medicine, University of Missouri, Columbia, MO 65212, USA; amoslandgrafj@missouri.edu; 3Department of Veterinary Medicine and Surgery, University of Missouri, Columbia, MO 65211, USA; 4Department of Veterinary Pathobiology, University of Missouri, Columbia, MO 65211, USA; 5Department of Pediatrics, School of Medicine, University of Missouri, Columbia, MO 65211, USA

**Keywords:** bioluminescence, luciferase, orthotopic, tumor, metastases, oncolytic adenovirus

## Abstract

Luciferase (luc) bioluminescence (BL) is the most used light-emitting protein that has been engineered to be expressed in multiple cancer cell lines, allowing for the detection of tumor nodules in vivo as it can penetrate most tissues. The goal of this study was to develop an oncolytic adenovirus (OAd)-resistant human triple-negative breast cancer (TNBC) that could express luciferase. Thus, when combining an OAd with chemotherapies or targeted therapies, we would be able to monitor the ability of these compounds to enhance OAd antitumor efficacy using BL in real time. The TNBC cell line HCC1937 was stably transfected with the plasmid pGL4.50[luc2/CMV/Hygro] (HCC1937/luc2). Once established, HCC1937/luc2 was orthotopically implanted in the 4th mammary gland fat pad of NSG (non-obese diabetic severe combined immunodeficiency disease gamma) female mice. Bioluminescence imaging (BLI) revealed that the HCC1937/luc2 cell line developed orthotopic breast tumor and lung metastasis over time. However, the integration of luc plasmid modified the HCC1937 phenotype, making HCC1937/luc2 more sensitive to OAdmCherry compared to the parental cell line and blunting the interferon (IFN) antiviral response. Testing two additional luc cell lines revealed that this was not a universal response; however, proper controls would need to be evaluated, as the integration of luciferase could affect the cells’ response to different treatments.

## 1. Introduction

The importance of accurately evaluating the efficacy of a preclinical anti-cancer therapy is critical for translational medicine, aiming to discover new drugs and therapies that can be used in the clinic. The most common assessment of the efficacy of such therapies is the measurement of subcutaneous (s.c.) tumor volume over time using standard caliper techniques and the comparison of tumor growth rates between untreated and treated mice groups. This is a low-cost, high-throughput method; however, caliper measurement is known to introduce bias, as the direction to measure the longest length is subjective, especially in tumors with complex morphologies [1]. Caliper contact with the tumor can also cause squeezing of the tumor, which leads to underestimation of the true volume [2]. Additionally, there is no standardized formula to measure the tumor volume, which can lead to non-reproducible results between different measurement techniques [3].

For many decades, the difficulty of accurately measuring the efficacy of therapies has limited many research groups to working with murine s.c. tumor models, as it can be easily measured using an external caliper. However, this model is not clinically relevant because it lacks the organ-specific conditions of the tumor microenvironment [4]. There are many cancer models that do not reproduce the clinical conditions of cancer development when using an s.c. tumor, such as the lung, pancreas, brain, and liver, among others. Therefore, the integration of orthotopic tumor models, especially when bioluminescence (BL) is incorporated to facilitate tumor detection, represents a powerful real-time tool to monitor cancer therapy efficacy. This overcomes the need for large cohorts with multiple time points, allowing for longitudinal monitoring and minimizing the number of mice needed.

The key advantages of bioluminescence imaging (BLI) include high sensitivity, resolution, and selectivity. As a result, this non-invasive imaging modality has been widely used for studying physiological, biochemical, and pathological processes in vivo [5]. In this regard, some notable applications of BLI include the labeling of cells, viruses, and bacteria, diagnosis and treatment of various diseases, assessment of protein-protein interactions, and study of gene expression [6]. In the case of cancer research, BLI offers accurate cancer detection, disease progression monitoring, the potential to detect metastatic nodules distant to the primary tumor site, and in vivo anti-tumor efficacy assessment [7].

The most common application of BLI in the biomedical field is the use of firefly luciferase (luc). The firefly luc gene was first cloned in 1985 [8]; since then, the luciferin–luciferase application has become the most reliable bioluminescent system for the emission of light and in vivo imaging [9]. Luciferin acts as the substrate and light-emitting molecule, whereas the luciferase enzyme oxidizes luciferin in the presence of oxygen and/or certain cofactors, such as calcium and magnesium [10]. The amount of light produced through this reaction is directly related to the amount of luciferin available for binding. Remarkably, the bioluminescence generated by the luciferin–luciferase system can penetrate both skin and tissues [11]. This characteristic allows researchers to incorporate the BL luciferin–luciferase system to assess physiological processes and diagnostics.

Although BLI has been widely implemented in multiple scientific fields, including cancer research, there are unresolved issues. For example, some scientists claim that luciferase expression may act as a tumor-associated antigen, creating an immune response against luciferase and altering tumor growth [12]. Similarly, there are reports that the luciferase plasmid internalization can alter the phenotype of the recipient cells [13]. This highlights the importance of evaluating whether the modified cell line expressing luciferase responds the same way as the parental cell line while being treated with a specific therapy. Therefore, the objectives of the present study are the development, establishment, and optimization of preclinically relevant orthotopic murine tumor models using in vivo BLI.

## 2. Results

To improve the detection threshold and reduce the subjectivity of manual tumor measurement, we integrated a BL expression system into a human triple-negative breast cancer (TNBC) cell line HCC1937/luc2, followed by in vivo imaging of the growth progression of the tumor and the development of metastases over time. Once the expression of luciferin was incorporated into the TNBC cell line, we tested whether the integration of the plasmid or the expression of luciferin affected the phenotype of the cell line to respond to an oncolytic adenovirus therapy (OAd), considering that any change in sensitivity would imply a more clinically relevant cancer model.

### 2.1. Establishment of a Bioluminescent TNBC Orthotopic Model

We stably transfected one human TNBC HCC1937 cell line and one mouse lung cancer CMT64 cell line for the expression of luciferase (HCC1937/luc2 and CMT64/luc2, respectively). We confirmed the production of BL signal in vitro for both cell lines (Appendix A). Furthermore, we compared Ad’s transduction and killing efficacy between CMT64 (parental) and CMT64/luc2 (modified). It was found that both cell lines were similarly infected by AdGFP and killed by OAdmCherry (Appendix A).

Next, we established an orthotopic model of TNBC by inoculating HCC1937/luc2 cells into the mammary fat pad of female NOD immune defective mice to allow the development of human-origin tumors. After 7 days post-inoculation, we observed a well-established tumor that emitted a colocalized BL signal (Figure 1A) that was homogeneous and consistent between all tested animals. Tumor growth was monitored 2 times per week. At day 14, a strong BL signal could be detected from the tumors (Figure 1B), and for both timeframes shown, the BL intensity was quantified, showing an increased signal detection corresponding to the change in tumor volume through the days with very low variability between individuals. This demonstrates that luciferase activity could be extrapolated to precisely follow tumor size, allowing for more accurate comparisons between groups (Figure 1C).

Once animals reached the experimental endpoint by reaching a tumor volume above 1000 mm^3^ on day 18, mice were scanned in vivo, detecting the presence of BL signal from the thorax (Figure 1D), corresponding to metastatic nodules allocated in the lungs (Figure 1E). The BL intensity detected in the in vivo scans highly correlated with the BL intensity observed in the ex vivo images (Pearson correlation coefficient of 0.79) (Figure 1F), validating the reliability of the luciferase BLI system to identify the presence of metastases in vivo corresponding to the location and intensity of the BLI detected directly from the isolated organs.

### 2.2. Luciferin Expression System Integration Modifying TNBC Cell Line Response to OAd Infection

After establishing an orthotopic TNBC model in which we could follow up on the tumor growth and the detection of metastases by BLI, we evaluated if this change in the cellular phenotype by expressing luciferase affected the sensitivity to treatment with OAds in vitro. First, we evaluated, the infectivity of replication-defective adenovirus expressing the green fluorescent protein (AdGFP) (control virus) by fluorescent microscopy, which had the capacity to bind to the coxsackie adenovirus receptor (CAR), enter the cell, and produce GFP that could be detected at the cellular cytoplasm (Figure 2A, top). Similarly, the replicative capacity of the OAd expressing the red fluorescent protein mCherry (OAdmCherry) was compared between both cell lines (Figure 2A, bottom). Increased sensitivity to being infected and a higher viral replication capacity were observed in the luciferase-expressing HCC1937/luc2 cell line.

We compared the killing effect of the OAd between the parental TNBC cell line HCC1937 and the modified cell line expressing luciferase, HCC1937/luc2. When treating the cells with the OAdmCherry after 72 h of infection, a significant decrease in viability was observed in the HCC1937/luc2 cell line (67% vs. 29%) compared with the parental HCC1937 (Figure 2B). We also tested a non-replicative, non-oncolytic AdGFP as a control for the presence of viral particles and did not observe any difference between cell lines.

This higher killing effect was associated with an increased replication, as confirmed by an increase of 30% in the number of copies detected in the OAdmCherry-treated HCC1937/luc2 supernatant (Figure 2C) and by western blot (WB) analysis with a higher expression of the adenovirus replication marker E1A, detecting a higher density of the protein in the HCC1937/luc2 cell line 24 h post-infection; this was especially noticeable at the lower molecular weight subunit of the protein (Figure 2D).

### 2.3. TNBC Cell Line Expressing Luciferin with a Decreased Antiviral Response

To understand the pathways that could be affected by the integration of the luciferase plasmid or the luciferase expression responsible for sensitization to the OAd infection on the modified HCC1937/luc2 cell line, we evaluated the expression levels of the main proteins regulating the cellular antiviral response.

We analyzed the protein expression profiles of the main regulators of the pathogen-activated immune response. This pathway was activated by interferon and included the interferon regulatory factor 9 (IRF9), the signal transducer, and the activator of transcription (STAT1 and STAT2); these proteins are responsible for the induction of interferon-stimulated genes (ISGs) [14]. Similarly, the oligoadenylate synthase family (OAS) are enzymes induced by interferon that recognized viral-produced RNA, inducing a potent antiviral activity [15]. We evaluated the expression levels of these proteins by WB, observing a sharp decline in the production of these proteins in the cells treated with the OAdmCherry, and this decrease was even more noticeable on the HCC1937/luc2 cell line (Figure 3) in accordance with the observed increased sensitivity to OAd. It was worth noting that an additional STAT1 band appeared when cells were infected by the OAdmCherry between the two reported subunits. These results altogether indicated that the antiviral response was downregulated when cells were infected with OAdmCherry, but it remained unaltered in the mock and AdGFP control groups. This effect was dose–response-dependent, as we observed less or no protein production in the highest concentration of OAds virus groups.

### 2.4. Detection of Lung Metastases in a Syngeneic Model of Orthotopic TNBC

Following the findings of the increased sensitivity to OAd infection of the modified TNBC cell line expressing luciferase, we tested the other commercially available TNBC cell line, 4T1/luc, and established a syngeneic orthotopic model of TNBC in BALB/C mice. The luciferase-expressing cell line 4T1/luc was inoculated in the mammary fat pad of female mice, and the tumoral growth was measured twice per week. Figure 4A,B show the BL-emitting tumors at 7 and 14 days, respectively. On day 7, tumors were barely palpable, but the BL signal was easily detectable and consistent across all the tumors. On day 14, tumors were well established, and the BL signal showed little variation between the individuals, adding to the replicability of the system to follow tumor growth quantified objectively by BL intensity. On day 20, tumors reached their maximum volumes for the experiment (above 1000 mm^3^), and the internal organs were isolated to identify the presence of metastases. We detected the presence of multiple metastatic nodules located in the lungs of all animals evaluated (Figure 4C). This method allowed the detection of tumor nodules in organs, even when it was not possible to identify visible or superficial metastases.

From the recovered lungs, we validated the presence of metastatic nodules deep in the tissue by H&E staining. The 4T1 TNBC model was highly invasive and could easily metastasize from the primary tumor site, with the lungs being one of the most frequently affected organs [16]. In Figure 4D, we show a comparison between the normal lung tissue structure and a metastatic nodule of 4T1 cells, with the characteristic morphologies of adenocarcinoma cells invading and thickening the alveolar walls. Tumor nodules were visible throughout the lung parenchyma, ranging in size and exhibiting irregular cell shapes. These tumor cells had larger nuclear sizes and irregular shapes than normal lung cells. Immunohistochemistry (IHC) from tumor-bearing lungs showed substantial Ki-67 positivity, indicating high tumor cell proliferation.

Additionally, we explored if the presence of the luciferase-expressing system changed the phenotype of these cells, affecting their response to OAds infection (Appendix A). We cultured the parental cell line 4T1 and the luciferase-expressing 4T1/luc in vitro and performed a dose–response assay with an increasing multiplicity of infection (MOI) of the virus. As observed in Figure 4E for the control AdGFP-treated group, no significant killing was observed, as this was a non-replicative virus. For the group infected with the OAdmCherry, there was a killing effect that peaked at 50% with the highest dose. However, no difference between the parental 4T1 and the 4T1/luc was observed. This indicated that for this cell line in particular, the inclusion of the luciferase expression did not affect the antiviral response mechanism, nor did it affect the capacity of the OAds to infect or replicate in these cells (Appendix A).

### 2.5. Luciferin Signal Allowing the Detection of In Vivo Tumors in a Metastatic Model of Lung Cancer

Evaluations of two orthotopic models of TNBC expressing luciferase demonstrated the use of BLI as a method to follow up the growth and identification of metastases in vivo and ex vivo. We were then interested in using the same detection system of BLI to monitor the establishment of orthotopic lung cancer models to monitor the presence of tumor infiltrates in vivo as a tool to test the efficacy of different treatments. This would represent a more clinically relevant model, detecting the tumor directly within the lungs by the quantification of BL.

First, we validated the capacity to produce BL when forming tumors in C56BL/6 mice by inoculating with the syngeneic lung cancer cell line TC-1/luc expressing luciferase. In Figure 5A, we detected the BL signal from well-established s.c. tumors after 14 days of TC-1/luc cells inoculation. After validating the detection of BLI from TC-1/luc tumors, we inoculated these cells by intravenous (i.v.) administration, which was previously described as an orthotopic lung tumor model [17]. After 16 days post i.v. inoculation, the lung tumors were well established, and the BL signal could be detected in vivo in all the mice (Figure 5B). After recovering the lungs from these animals, we could verify the presence of several tumor nodules in the lungs (Figure 5C), and their ex vivo BL intensity signal corresponded to the level of intensity observed in the in vivo imaging. This result further supported the reliability of the measures obtained in the live animal corresponding with the actual size and extent of the tumor infiltrates.

From the obtained lungs, H&E staining was performed to corroborate that the BL signal we obtained corresponded to actual tumor infiltrations. Figure 5D shows evidence of tumor invasion into surrounding lung structures, including the alveoli and small blood vessels. Ki-67 staining indicated a strong positivity in most of the nuclei from the denser area corresponding to a tumor nodule. There was a clear difference between the normal lung tissue and the dense tumor nodule from tumor-bearing lungs, confirming the establishment of an orthotopic lung cancer model that could be monitored by BLI in vivo.

Lastly, we evaluated the response to OAd infection between the parental cell line TC-1 and the luciferase-expressing TC-1/luc, and there was not a significant difference in the capacity to be infected between these two cell lines (Appendix A). Viability between both cell lines showed a significant difference only at the highest OAdmCherry dose (Figure 5E). However, due to its high resistance to the killing effect of this specific OAd, the antiviral mechanism was not further evaluated.

## 3. Discussion

As science progresses toward discoveries of new generations of drugs and technologies offering new and safer alternatives to fight cancer, more accurate research tools to evaluate treatment effectiveness are needed. The use of calipers to manually register the size of tumors to evaluate differences between treatment groups has been the standard for most laboratories around the world; however, this method is highly subjective and presents several limitations, such as the difficulty of measuring the depth of the tumor and the impossibility to quantify any form of internal tumors. In a study by Brough et al. [18], they evaluated the variability between operators to assign animals in different groups according to the measure obtained via calipers. They observed that a different operator had a 59% probability of reassigning an animal to a different group. This highlights the need to provide tools and methodologies that can overcome the issues and biases inherent to the subjective use of calipers as a therapy efficacy measurement standard. In this study, we propose the use of optical imaging and objective quantification of emitted light as an alternative measurement technique to improve accuracy by comparing tumor sizes between individuals and groups.

Optical imaging, including fluorescence and bioluminescence, is the most popular method for in vivo imaging in mice. Bioluminescence imaging is considered to be superior to fluorescence imaging owing to the lack of both autofluorescence and the scattering of excitation light [19]. Many laboratories will often engineer cell lines and primary cultures to stably express one of these luciferase genes for cancer research. This is often achieved through the transfection of a DNA plasmid construct or the infection of the cells with lentiviral vectors. There are research teams dedicated to producing biologically modified versions of different BL probes to increase the signal outcome and improve the sensitivity of imaging [20].

Bioluminescence imaging is a non-invasive optical imaging modality designed to visualize and quantify bioluminescent signals in tissues and represents a powerful tool to efficiently monitor cancer therapy efficacy [21]. BLI is based on the detection of visible light produced during the enzyme-mediated oxidation of a substrate when the enzyme is expressed in vivo as a molecular reporter. The emitted light signal is then detected through in vivo imaging systems and subsequently quantified to reflect the overall tumor size. Researchers can use this information to determine when treatments should be initiated and when the tumor burden becomes too excessive for the animal or to monitor the efficacy of various treatments longitudinally.

The injection of luciferase-expressing cancer cells into various animal models is one of the most common ways in which bioluminescence is utilized in cancer research. The type of cancer being studied will often determine the inoculation modality needed for a given animal model. Glioblastoma research, for example, will typically require the intracranial injection of luciferase-expressing cells [22]; a lung cancer study will require a subcutaneous, intravenous, or even intrathoracic injection of luciferase-expressing cancer cells [23]. Regardless of what injection approach is utilized, researchers can monitor the progression of any given cancer by subsequently administering luciferin to the treated animals. The luciferin will quickly bind to the luciferase enzyme expressed within the cancer cells, thus leading to the chemical reaction that produces bioluminescence.

Following the versatility of using BLI to track the presence of cancer cells and the formation of superficial or deep tumors, we modified two cancer cell lines, the human TNBC HCC1937 and the mouse lung cancer CMT64, to be able to study in vivo. After the luciferase plasmid was incorporated into both cell lines, we verified the production of BL in vitro (Appendix A) before the initiation of the multiple experiments detailed in this study.

We focused on developing an orthotopic model for evaluating therapy efficacy to study more clinically relevant breast and lung cancer models. Triple-negative breast cancer (TNBC) constitutes 15–20% of breast cancer; it is associated with a younger age and advanced stage at diagnosis, higher risk of visceral metastasis, and poorer outcomes [24]. Similarly, lung cancer is the leading cause of cancer-related deaths in the world [25]. The higher the sensitivity of the detection method is, the higher the probability of detecting these cancers at an early stage, greatly improving long-term survival. In our study, we achieved higher levels of sensitivity by using BLI as a tracking tool to detect the presence of primary tumors and metastatic nodules in anesthetized animals in vivo.

Recent advances in the development of the latest generation of chemotherapy and targeted immunotherapies have significantly impacted the field of cancer research. These novel therapies and potential new drugs require a reliable, robust, and reproducible method to evaluate their anti-tumor efficacy in a more precise and rigorous manner. We evaluated the use of oncolytic virus as a promising direction for TNBC and lung cancer research.

In this study, we tested two cancer cell lines modified to express luciferase in our laboratory (HCC1937/luc2 and CMT64/luc2) and two commercially available cancer cell lines expressing luciferase (4T1/luc and TC-1/luc). We confirmed the production of BL signal in vitro (Appendix A) and compared the sensitivity and killing effect of our OAdmCherry therapy to infect and replicate in these cancer cell lines (Appendix A). As CMT64/luc2 showed a poor response to the OAdmCherry infection (Appendix A), we decided not to use this cell line in the in vivo mice model.

Surprisingly, the incorporation of the luc2 plasmid into the HCC1937 cell line conferred a higher capacity to be infected by AdGFP and OAdmCherry. This change in phenotype was confirmed to be associated with a downregulation of the antiviral response pathway. Interestingly, HCC1937 has been previously described as a highly heterogeneous TNBC cell line because it carries a mutated BRCA1 gene, resulting in a defective DNA repair response [26]. In an attempt to categorize different TNBC cell lines according to their tumorigenesis and clinical relevance, Lehmann et al. described HCC1937 as a basal-like 1 breast cancer associated with genomic instability [27]. These mutagenic characteristics of the cell line HCC1937 make it particularly sensitive to altering its phenotype when exposed to genetic stress, such as the incorporation of the luc2 plasmid. Due to the diversity in heterogeneity between the several models of cancer and available cancer cell lines, it is advised to always keep in mind the possibility that altering a cell line to express luciferase or other protein can impact its phenotype and response to specific treatments.

We previously explored the efficacy of OAds to TC-1 lung cancer tumors, obtaining promising results by greatly decreasing tumor development [28]. However, we want to further explore the use of the OAd therapy by in-depth following up on tumor growth and detection of metastases using a clinically relevant cancer model. Hence, the aim of this study was to implement orthotopic cancer models equipped with the BL detection system for TNBC, HCC1937/luc2 and 4T1/luc, and lung cancer model TC-1/luc.

## 4. Materials and Methods

### 4.1. Cell Lines

The human TNBC cell line HCC1937 (Cat. CRL-2336, ATCC, Manassas, VA, USA) and murine lung adenocarcinoma cell line CMT64 (Cat. 10032301, Millipore-Sigma, Burlington, MA, USA) expressing luciferase (HCC1937/luc2 and CMT64/luc2, respectively) were generated in our laboratory, as described in the luciferase plasmid transfection methodology. The 4T1/luc2 was a murine triple-negative breast cancer (TNBC) luciferase-expressing cell line derived from parental line 4T1 by transduction with lentiviral vector encoding firefly luciferase gene (luc2) under the control of the EF-1 alpha promoter, and it was provided by Dr. Cristian Rodriguez-Aguayo from the MD Anderson Cancer Center. The TC-1/luc is a murine lung cancer cell line derived from primary epithelial cells of C57BL/6 mice co-transformed with HPV-16 E6/E7 and c-Ha-ras oncogenes, and expressing luciferase was provided by Dr. T. C. Wu from Johns Hopkins University.

### 4.2. Hygromycin Killing Dose-Response Assay

To prepare the parental cell lines HCC1937 and CMT64 to incorporate the luciferase-expressing plasmid, it was necessary to determine their natural sensitivity to hygromycin, as this was the antibiotic resistance encoded in the luciferase plasmid. Cells were plated at a density of 2 × 10^4^ cells/well in a 24-well plate. On the following day, hygromycin (Cat. H0654, Millipore-Sigma, Burlington, MA, USA) was added to the cells at increasing concentrations (0, 10, 50, 100, 200, 400, 600, and 800 µg/mL) cells were incubated for three days. At 72 h post-treatment with hygromycin, cell viability was evaluated by alamarBlue (Cat. A50101, Thermo Fisher Scientific, Waltham, MA, USA) according to the manufacturer’s instructions.

### 4.3. Plasmids and Generation of Stably Transfected Luciferase-Expressing Cell Lines

Once we determined that hygromycin at a concentration of 800 µg/mL resulted in a cell viability of less than 20%, we proceeded to perform a stable transfection. HCC1937 or CMT64 cell lines were seeded at 1.5 × 10^5^ cells per well in a 24-well plate, followed by transfection with the plasmid pGL4.50[luc2/CMV/Hygro] (Cat. E1310, Promega, Madison, WI, USA). Transfection was performed with jetPRIME transfection reagent (Polyplus-transfection, Illkirch, France) according to the manufacturer’s instructions. At 48 h post-transfection, cells were trypsinized, serially diluted, and cultured with a fresh medium supplement with hygromycin at a concentration of 800 µg/mL. The medium containing hygromycin was changed every three days for up to 14 days until small cell colonies appeared. Single-cell colonies were expanded and cultured in the presence of hygromycin.

### 4.4. Evaluation of Luciferase Expression from Stably Transfected Cell Lines

Once the stably transfected cell lines were established, the expression of luciferase was evaluated in vitro via BLI using the advanced molecular imager (AMI-HTX, Spectral Instruments Imaging, Tucson, AZ, USA). The cell lines HCC1937 or CMT64 were seeded at increasing cell densities of 1 × 10^5^, 3 × 10^5^, and 9 × 10^5^ cells per well in 6-well plate. Then, 24 h later, D-luciferin (Cat. 122799, Revvity, Waltham, MA, USA) was added to the cells at a concentration of 15 µg/mL and incubated for 10 min, which was followed by BLI in vitro using AMI-HTX imaging system. Once the positive expression of luciferase was confirmed, the cell lines were renamed as follows: HCC1937/luc2 and CMT64/luc2.

### 4.5. Orthotopic Inoculation of Human or Syngeneic Triple-Negative Breast Cancer Mouse Tumor Model and Bioluminescence Imaging

Human or syngeneic orthotopic TNBC tumors were formed into the fourth mammary gland fat pad of 6-week-old NSG (NOD scid gamma) or Balb/c female mice, respectively, as follows: mouse under isoflurane (2%) anesthesia were injected into the fourth mammary gland fat pad with 1 × 10^6^/100 µL of human HCC1937/luc2 cells in phosphate-buffered saline (PBS) or 2 × 10^5^/100 µL PBS of 4T1/luc2 cells for syngeneic model. Animals were housed in a barrier animal facility at the University of Missouri–Columbia in accordance with National Institute of Health (NIH) guidelines (Guide for the Care and Use of Laboratory Animals, NIH Publication No. 8023, Revised 1978). All procedures were analyzed and approved by the University of Missouri Institutional Animal Care and Use Committee (IACUC) (protocol no. 23140).

Tumor growth was monitored twice per week by BLI measuring via AMI-HTX as follows: mice were injected intraperitoneally with 100 µL of D-luciferin at 15 mg/mL, which was followed by placing the animals into an anesthesia induction chamber and anesthetized using isoflurane (Dechra, Overland Park, KS, USA). Then, it was placed in the AMI-HTX imaging system for whole body scanning using BLI.

### 4.6. Inoculation of Subcutaneous Tumors and Detection by BLI

Male (6- to 8-week-old) C57BL/6 mice were purchased from the Jackson Laboratory (Bar Harbor, ME, USA). Subcutaneous (s.c.) tumors were formed in the flanks of 6-week-old C57BL/6 mice by injecting 1 × 10^5^ TC-1/luc cells in 100 µL of PBS. Tumor growth was monitored twice per week by BLI via AMI-HTX, as described above.

### 4.7. Establishment of the Orthotopic Lung Cancer Model and Bioluminescence Imaging

For the orthotopic lung tumors model, 6-week-old C57BL/6 mice were injected with 1 × 10^5^ TC-1/luc in 100 µL of PBS 1X into the lateral tail vein. Tumor growth was monitored twice per week by BLI via AMI-HTX. Mice were injected intraperitoneally with D-luciferin, as detailed above, followed by placing the animals into an anesthesia induction chamber and anesthetized using isoflurane. It was placed in the AMI-HTX imaging system for whole body scanning using BLI.

### 4.8. In Vitro Cell Viability Assay

Cell lines were cultured in 24-well plates at a density of 2 × 10^4^ and infected the next day with AdGFP or OAdmCherry in a dose–response assay from an MOI 0 to 200 and incubated for 72 h to observe the killing effect. After the elapsed time, treatment groups were incubated with AlamarBlue (Cat. A50101, Thermo Fisher Scientific) and supernatant, and they were read using fluorescence in a plate reader (Synergy H1, BioTek, Winooski, VT, USA) to determine the metabolic activity of the cells and to extrapolate cell viability.

### 4.9. Western Blots

Whole protein lysates were obtained from cells after treatment using RIPA buffer with protease (Cat. 78429, Thermo Fisher) and fosfatase (Cat. P2850, Millipore-Sigma, Burlington, MA, USA) inhibitors. Total proteins were quantified using the PIERCE BCA Protein Assay Kit (Cat. 23225, Thermo Fisher). For electrophoresis, SDS–polyacrylamide gels at 10% were cast, and the protein samples were loaded. Proteins were transferred to a PVDF membrane (Cat. 10600023, GE Healthcare Life Sciences, Pittsburgh, PA, USA). The following primary antibodies were used: anti-E1A (Cat. 554155, BD Pharmigen, San Diego, CA, USA), anti-IRF9 (Cat. 76684, Cell Signaling, Danvers, MA, USA), anti-STAT1 (Cat. 14994, Cell Signaling), anti-STAT2 (Cat. 72604, Cell Signaling), anti- OAS1 (Cat. 14498, Cell Signaling), and anti-actin (Cat. A2066, Millipore-Sigma).

### 4.10. qPCR for Adenovirus Replication Assay

Replication of the OAdmCherry after infecting the cell line HCC1937/luc2 was determined by detection of the viral particles released to the supernatant 24 and 48 h post-infection. The viral genome was purified using a NucleoSpin virus column (Cat. 740983, Takara, San Jose, CA, USA), and the viral copy number was quantified using the Adeno-X qPCR Titration Kit (Cat. 632252, Takara).

### 4.11. Hematoxilin and Eosin Staining

Normal tissues and tumor tissues collected from mice lungs and colons were fixed in 10% buffered formalin. The materials were processed, encased in paraffin blocks, and cut into 4-μm-thick slices. The tissue slices from all treatment groups were stained with hematoxylin and eosin (H&E) and trichrome. The 4-μm sections were mounted on glass slides, deparaffinized in xylene, and rehydrated using graded ethanol solutions. Hematoxylin was added to the cell nuclei for 5–10 min and then rinsed to eliminate excess stain. Eosin was used for 1–2 min to stain the cytoplasmic and extracellular matrices. Following staining, the slides were dehydrated, cleaned in xylene, and mounted with coverslips using an appropriate mounting medium. Tumor morphology, cellularity, and necrosis were evaluated histologically using a light microscope at 20× or 40× magnification. Representative pictures were collected for analysis.

### 4.12. Immunohistochemistry

Tissue sections were baked overnight, after which they were deparaffinized in xylene twice for 5 min each. The sections were then rehydrated using graded alcohol solutions. Antigen retrieval was carried out using a citrate buffer (0.01 M, 95 °C, pH 6.0). Endogenous peroxidase activity was inhibited by 3% H_2_O_2_ in methanol for an hour. After washing with PBS, the sections were blocked with 2.5% horse serum (ImmPRESS kit; Vector Labs, Burlingame, CA, USA) for 2 h. The sections were then treated with primary antibodies overnight at 4 °C. After washing with PBS containing 0.01% Tween 20 (PBST), the sections were treated for 30 min with a minantibody (peroxidase-labeled anti-mouse/anti-rabbit IgG; ImmPRESS kit, Vector Labs). The sections were then washed three times with PBST for five minutes each and developed with a DAB substrate kit (Vector Labs). Hematoxylin was used to counterstain the specimens. Before imaging, the slides were dehydrated using graded alcohol solutions, cleaned in xylene twice for 5 min each, and mounted with toluene [29]. The anti-Ki67 antibody (Cat. ab15580, Abcam, Cambridge, MA, USA) was utilized in the immunohistochemistry study.

### 4.13. Statistical Analysis

All data were analyzed using GraphPad Prism v10.3 (GraphPad Software, Inc., La Jolla, CA, USA). For statistical differences between groups a two-way ANOVA with Tukey was performed for multiple comparisons. For correlation analysis, a Pearson correlation coefficient test was performed.

## 5. Conclusions

In this study, we explored the use of BLI for the detection of primary tumors and metastatic nodules in vivo, using a luciferase signal as a more accurate and early-stage detection tool for evaluating the responses of anti-cancer therapies in TNBC and lung cancer orthotopic mouse models.

The TNBC cell line, HCC1937/luc2, decreased its antiviral response phenotype after the luciferase plasmid was incorporated, gaining a higher sensitivity to infection, replication, and killing by the OAdmCherry.

We successfully established BL-responding orthotopic cancer models from a TNBC of human origin, HCC1937/luc2, and a mouse syngeneic TNBC model, 4T1/luc. For the lung cancer model, we evaluated the cell line TC-1/luc. We could detect the development of tumors in the primary site of inoculation and/or metastatic nodules at the lungs that were easily identified by the use of BLI. We confirmed the presence of tumors by histological analyses of the tissue structure and proliferation marker Ki-67.

## Figures and Tables

**Figure 1 ijms-25-10418-f001:**
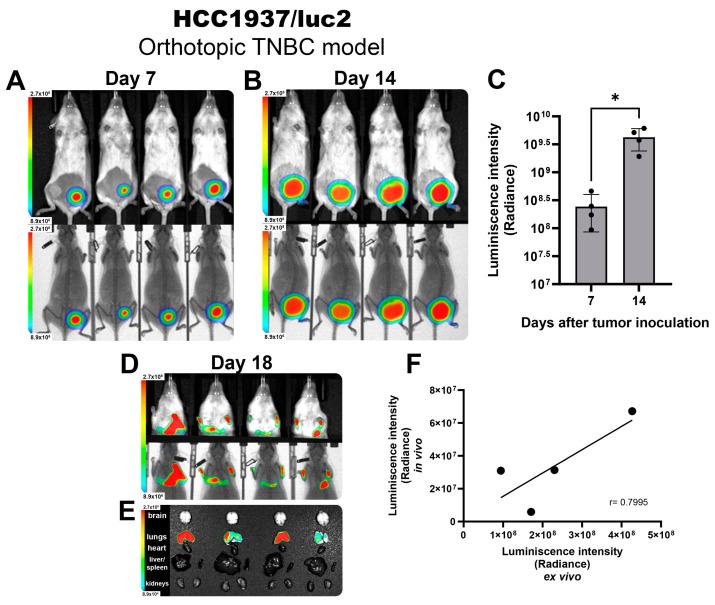
HCC1937/luc2 TNBC tumors and metastatic nodules detected by BLI in vivo imaging. Images showing the intensity of BLI from tumors corresponding to 7 (**A**) and 14 days (**B**) post-inoculation. Photography and x-ray. (**C**) BLI quantification allows for an accurate tumor size comparison. (**D**) Detection of BLI from lung metastases in vivo and ex vivo (**E**) and their signal quantification confirming a positive correlation of the intensity observed (**F**). The graphs show 3 independent experiments (Mean ± SD, * *p* < 0.05) for linear regression (r = Pearson correlation coefficient).

**Figure 2 ijms-25-10418-f002:**
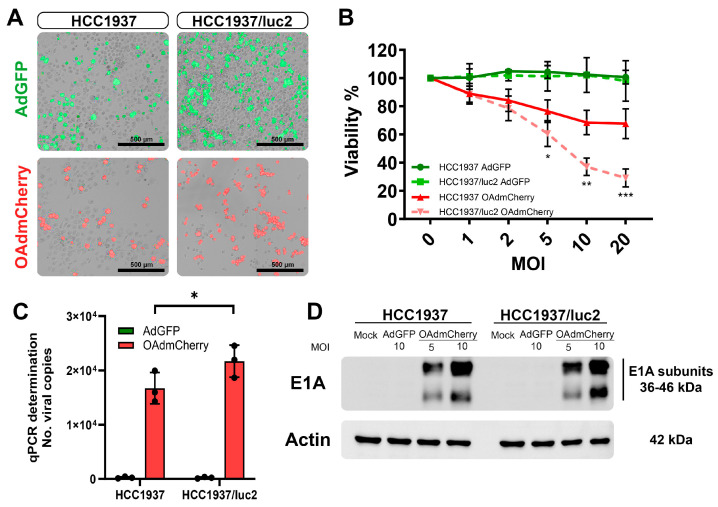
Incorporation of the luciferase expression system increases the infectivity and killing effect of OAdmCherry in the TNBC HCC1937/luc2 cell line. (**A**) Infectivity by AdGFP (top) and replication by OAdmCherry (bottom) are increased in the HCC1937/luc2 cell line at 48 h. (**B**) Viability is decreased at 72 h in the OAdmCherry infected HCC1937/luc2. (**C**) The number of Adenovirus DNA copies by qPCR (**D**) and expression of the E1A adenovirus replication marker is increased in the HCC1937/luc2 cells after 24 h of infection. The graphs show 3 independent experiments (Mean ± SD, * *p* < 0.05, ** *p* < 0.01, *** *p* < 0.001).

**Figure 3 ijms-25-10418-f003:**
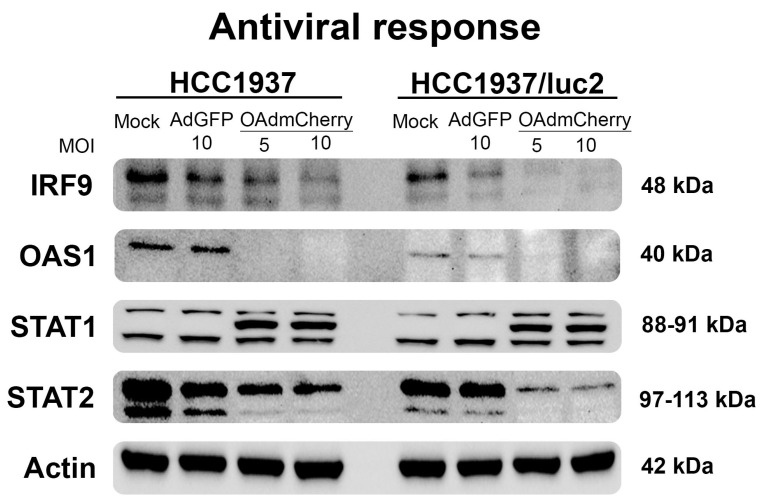
The antiviral response is further downregulated after OAdmCherry infection in the HCC1937/luc2 cell line. WB shows the main proteins of the interferon-activated immunity. Infection with OAdmCherry decreases the production of these proteins in a dose–response manner.

**Figure 4 ijms-25-10418-f004:**
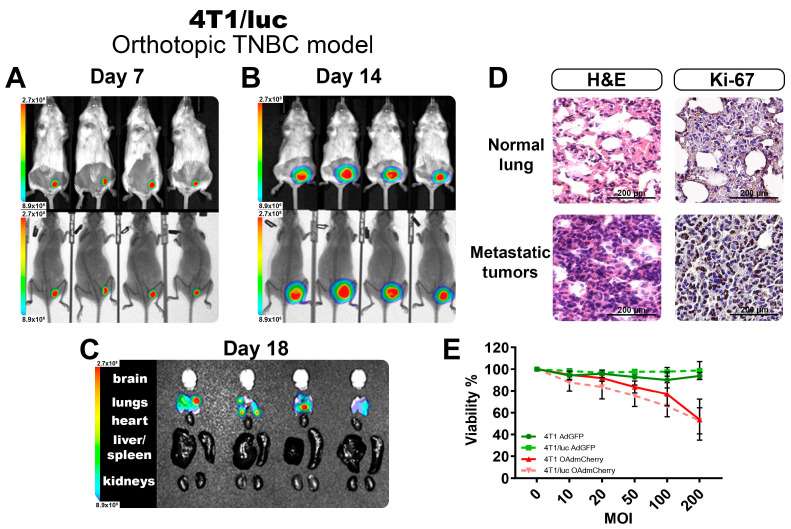
Evaluation of the TNBC 4T1/luc orthotopic model. Monitoring of the tumor growth at day 7 (**A**) and day 14 (**B**), as well as the identification of metastatic nodules in organs (**C**) by BLI imaging. (**D**) The normal tissues and tumor tissues recovered from the lungs demonstrate the presence of metastatic infiltrates. (**E**) Viability assay after 72 h of OAd infection in both 4T1 and 4T1/luc cell lines. The graphs show 3 independent experiments (Mean ± SD, no significance NS).

**Figure 5 ijms-25-10418-f005:**
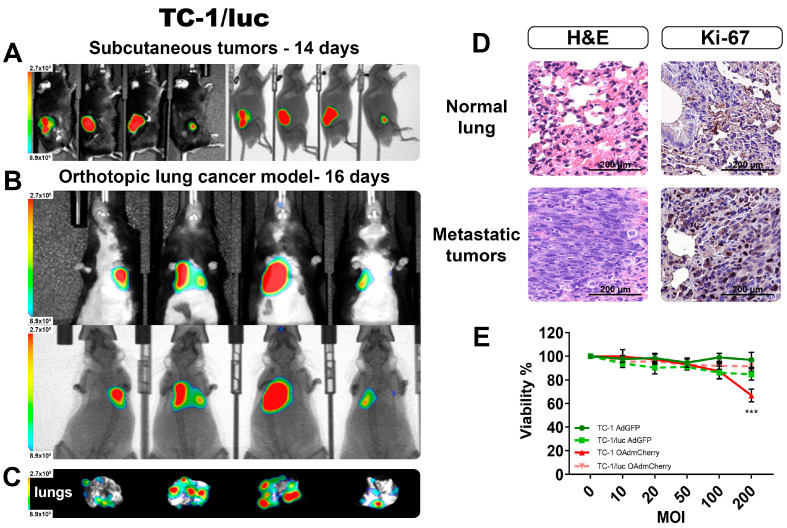
Monitoring by the BLI of an orthotopic TC-1/luc lung cancer model. (**A**) Subcutaneous tumors confirm the emission of BLI from the TC-1/luc cell line. (**B**) The detection of orthotopic lung tumors in vivo BLI imaging, and (**C**) the corresponding ex vivo lungs showing the presence of several tumoral nodules. (**D**) H&E and IHC staining confirmed the presence of tumor infiltrates and increased proliferation in the lungs. (**E**) Viability comparison between TC-1 and TC-1/luc after 72 h infection with OAds. The graphs show 3 independent experiments (Mean ± SD, *** *p* < 0.001).

## Data Availability

No data set was generated in the present study. All current data are presented in the above figures. Experimental data are available upon reasonable request.

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
