# Peer review of "Establishment of Translational Luciferase-Based Cancer Models to Evaluate Antitumoral Therapies"

_ijms, 2024, doi:10.3390/ijms251910418_

Round 1
Reviewer 1 Report
Comments and Suggestions for Authors
In this manuscript, the authors characterized cell lines expressing firefly luciferase (luc2) for potential applications of antitumoral therapies. The study addresses an important topic, particularly the concern that reporter genes may alter cell properties -an issue encountered by many researchers. Overall, the datasets presented in the manuscript are well-organized and contain meaningful findings, demonstrating changes in immunoreactivity across multiple human and murine cell lines. The results underscore the importance of meticulous characterization of cells when reporter genes are utilized. Therefore, the reviewer recommends the publication of this paper in International Journal of Molecular Science, provided that the authors reasonably address points outlined below.
1. I could not find the Fig.S1B/C/D in the results section. It’s better to mention these data in the results section.
2. According to the method described, the authors expanded single colonies to obtain HCC1937/luc cell line. HCC1937/luc2 exhibited greater sensitivity to OAdmChery treatment compared to the parental cell line. Do the authors have any discussion on whether this change could be attributed cellular heterogeneity within the parental cell population rather than the genetic transduction of luc2 ?
3. The scale value of the pseudo color bar in bioluminescence images (Fig.1A/B/D/E, Fig.4A/B/C and Fig.5A/B/C) are not readable. Please provide them with bigger font size.
4. It is recommended that the unit of scale bars in the microscopic images is presented in μm rather than μM.

Author Response
Comment: I could not find the Fig.S1B/C/D in the results section. It’s better to mention these data in the results section.
Response: Figure S1 corresponding to the characterization of the BL and infectivity by OAd properties of the lung cancer CMT64/luc2, is now properly mentioned in both results and discussion sections. (page 2, lines 91 – 94 and page 9, lines 331-334).
Comment: According to the method described, the authors expanded single colonies to obtain HCC1937/luc cell line. HCC1937/luc2 exhibited greater sensitivity to OAdmChery treatment compared to the parental cell line. Do the authors have any discussion on whether this change could be attributed cellular heterogeneity within the parental cell population rather than the genetic transduction of luc2?
Response: We have expanded the discussion where we mention previously described characteristics of the TNBC cell line HCC1937, like a mutated BRCA1 gene that results in a defective DNA repair response [1], as well as a high tumor heterogeneity showing a basal-like 1 phenotype for TNBC that can be associated with genomic instability [2]. These properties of the parental cell line make these cells more sensitive to change its phenotype when exposed to a mutagenic stress, as the incorporation of the luc2 plasmid. (Page 9, lines 336 - 345 and page 10, lines 346 - 349). We also added these two new references.
Chen, J., et al., Stable Interaction between the Products of the BRCA1 and BRCA2 Tumor Suppressor Genes in Mitotic and Meiotic Cells. Molecular Cell, 1998. 2(3): p. 317-328.
Lehmann, B.D., et al., Identification of human triple-negative breast cancer subtypes and preclinical models for selection of targeted therapies. Journal of Clinical Investigation, 2011. 121(7): p. 2750-2767.
Comments: The scale value of the pseudo color bar in bioluminescence images (Fig.1A/B/D/E, Fig.4A/B/C and Fig.5A/B/C) are not readable. Please provide them with bigger font size.
Response: We increased the size of the texts indicating the value of the scale bar to make it easier to read in all the figures.
Comment: It is recommended that the unit of scale bars in the microscopic images is presented in μm rather than μM.
Response: We appreciate the attention to this detail. It was an imprecision on our side that has been corrected accordingly in all the corresponding figures.
Reviewer 2 Report
Comments and Suggestions for Authors
The present work involved the evaluation of two cancer cell lines, namely HCC1937/luc2 and CMT64/luc2, which were genetically altered to express luciferase. Additionally, two commercially available cancer cell lines, namely 4T1/luc and TC-1/luc, were also evaluated. In vitro, they verified the generation of BL signal and conducted a comparative analysis of the sensitivity and lethality of our OAdmCherry treatment in infecting and replicating in these selected cancer cell lines. Given the inadequate response of CMT64/luc2 to the OAdmCherry infection, the decision was made to exclude this cell line from the in vivo mouse model.
A BL reporting orthotopic cancer model was successfully developed using human TNBC, HCC1937/luc2, and a mouse syngeneic TNBC model, 4T1/luc. For the lung cancer model, the cell line TC-1/luc was explored. Tumours could be detected at the initial site of inoculation and/or metastatic nodules in the lungs, which were readily identified by BLI and verified by histological examination of tissue structure and proliferation marker Ki-67.
It is a very interesting, well-designed, and well-constructed work. Valuable results will be added to the cell culture experiments, which could serve as a basis for further experiments in the future.
The methodologies are modern, and the statistical analyses are correct.
Images and graphs are illustrative and aid understanding.
However, the discussion and the conclusion are not a single unit. I feel that the discussion is incomplete. The results obtained should be discussed in more detail, and possible biasing factors should be analyzed.
In the experiment, it was found that the TNBC cell line HCC1937/luc2 had a reduced antiviral response phenotype after incorporation of the Luciferase plasmid and gained greater susceptibility to infection, replication, and killing by OAdmCherry.
This highlights an interesting phenomenon. The modification of cancer cells with luciferase plasmid affects the biological response of cancer cells against other viruses. Given this, how much of a bias might modification with luciferase plasmids have for further experiments?
Further exploration of this aspect would also be important.
I suggest a major revision.
Author Response
Comment: The present work involved the evaluation of two cancer cell lines, namely HCC1937/luc2 and CMT64/luc2, which were genetically altered to express luciferase. Additionally, two commercially available cancer cell lines, namely 4T1/luc and TC-1/luc, were also evaluated. In vitro, they verified the generation of BL signal and conducted a comparative analysis of the sensitivity and lethality of our OAdmCherry treatment in infecting and replicating in these selected cancer cell lines. Given the inadequate response of CMT64/luc2 to the OAdmCherry infection, the decision was made to exclude this cell line from the in vivo mouse model.
Response: We found that integration of Luciferase plasmid into CMT64 (CMT64/Luc2) did not alter its response to OAdmCherry killing effect when compared with parental CMT64 cell line (Supplementary figure 1D). Both cell lines displayed suboptimal response to OAdmCherry-mediated oncolytic cell death (Supplementary figure 1D).
Comment: A BL reporting orthotopic cancer model was successfully developed using human TNBC, HCC1937/luc2, and a mouse syngeneic TNBC model, 4T1/luc. For the lung cancer model, the cell line TC-1/luc was explored. Tumours could be detected at the initial site of inoculation and/or metastatic nodules in the lungs, which were readily identified by BLI and verified by histological examination of tissue structure and proliferation marker Ki-67.
Response: This study represents an additional effort to demonstrate the feasibility to use bioluminescence imaging to monitoring tumor burden. However, capital equipment invest may be required for the use of bioluminescence imaging in vivo.
Comment: It is a very interesting, well-designed, and well-constructed work. Valuable results will be added to the cell culture experiments, which could serve as a basis for further experiments in the future.
Response: Thank you for your enthusiasm in our study
Comment: The methodologies are modern, and the statistical analyses are correct.
Images and graphs are illustrative and aid understanding.
Response: We greatly appreciate the reviewer’s encouraging comments, and the dedication put into so accurately describing our proposed models in this study.
Comment: However, the discussion and the conclusion are not a single unit. I feel that the discussion is incomplete. The results obtained should be discussed in more detail, and possible biasing factors should be analyzed.
Response: The discussion has been extended to explain more in-depth the mechanisms that confer phenotypical changes in the TNBC cell line studied (HCC1937). The conclusions section can be found in a separate section on pages 12 and 13, lines 494-507.
Comment: In the experiment, it was found that the TNBC cell line HCC1937/luc2 had a reduced antiviral response phenotype after incorporation of the Luciferase plasmid and gained greater susceptibility to infection, replication, and killing by OAdmCherry. This highlights an interesting phenomenon. The modification of cancer cells with luciferase plasmid affects the biological response of cancer cells against other viruses. Given this, how much of a bias might modification with luciferase plasmids have for further experiments? Further exploration of this aspect would also be important.
Response: This is an interesting and appropriate request. We have incorporated into the discussion a description of the characteristics of the HCC1937 cell line that make it genotypically unstable and a target to phenotypic alterations after incorporating the luc2 plasmid. Additionally, we added a mention of how the expression of luciferase, or other proteins can be a source of variations in the response to treatments. (Page 9, lines 332-345).
Comment: I suggest a major revision.
Response: Figures have been modified to a higher resolution and the scales of BL are now easier to read. Supplementary data is now mentioned in more detail in the results and discussion section.
Round 2
Reviewer 2 Report
Comments and Suggestions for Authors
The authors corrected the manuscript according to the previous suggestions. The revised version is now suitable for publication. Congratulations!